# Reassessing the modularity of gene co-expression networks using the Stochastic Block Model

Diogo Melo[1,2¤]*, Luisa F. Pallares[1,2,3], Julien F. Ayroles[1,2]*

**1** Lewis-Sigler Institute for Integrative Genomics, Princeton University, Princeton, New Jersey, United States of America, **2** Department of Ecology and Evolutionary Biology, Princeton University, Princeton, New Jersey, United States of America, **3** Friedrich Miescher Laboratory of the Max Planck Society, Tübingen, Germany

¤ Current address: Departamento de Genética e Biologia Evolutiva, Instituto de Biociências, Universidade de São Paulo, São Paulo, SP, Brasil

* damelo@princeton.edu (DM); jayroles@princeton.edu (JFA)

**Data Availability Statement:** All relevant data are within the paper and its Supporting Information files. Code for using graph-tools to cluster

## Abstract

Finding communities in gene co-expression networks is a common first step toward extracting biological insight from these complex datasets. Most community detection algorithms expect genes to be organized into assortative modules, that is, groups of genes that are more associated with each other than with genes in other groups. While it is reasonable to expect that these modules exist, using methods that assume they exist a priori is risky, as it guarantees that alternative organizations of gene interactions will be ignored. Here, we ask: can we find meaningful communities without imposing a modular organization on gene co-expression networks, and how modular are these communities? For this, we use a recently developed community detection method, the weighted degree corrected stochastic block model (SBM), that does not assume that assortative modules exist. Instead, the SBM attempts to efficiently use all information contained in the co-expression network to separate the genes into hierarchically organized blocks of genes. Using RNAseq gene expression data measured in two tissues derived from an outbred population of Drosophila melanogaster, we show that (a) the SBM is able to find ten times as many groups as competing methods, that (b) several of those gene groups are not modular, and that (c) the functional enrichment for non-modular groups is as strong as for modular communities. These results show that the transcriptome is structured in more complex ways than traditionally thought and that we should revisit the long-standing assumption that modularity is the main driver of the structuring of gene co-expression networks.

## Author summary

Understanding how genes work together is crucial for unraveling the biological processes underlying complex traits. To gain insight into these genetic interactions, researchers often analyze gene co-expression networks, in which genes are linked based on the similarity of their expression patterns among different individuals. Traditionally, it has been

expression data using the SBM can be found at https://github.com/ayroles-lab/SBM-tools.

**Funding:** D.M. is funded by a fellowship from the Princeton Presidential Postdoctoral Research Fellows Program. L.P. was funded by a Long-Term Postdoctoral Fellowship from the Human Frontiers Science Program and is funded by the Max Planck Society. J.A. is funded by grants from the NIH: National Institute of Environmental Health Sciences (R01-ES029929) and National Institute of General Medical Sciences (NIGMS) (R35GM124881). The funders had no role in study design, data collection and analysis, decision to publish, or preparation of the manuscript.

**Competing interests:** The authors have declared that no competing interests exist.

assumed that these networks are organized into distinct assortative modules, in which genes are more connected to each other within modules than between them. However, by using a novel non-parametric clustering approach called the Stochastic Block Model, we show that fruit fly transcriptomes contain not only assortative, modular gene clusters, but also functionally relevant non-modular clusters that would be overlooked by standard methods. This suggests that transcriptional networks may be more complex and diverse than previously thought. Our findings highlight the importance of using unbiased clustering techniques to fully capture the various architectures of gene co-expression networks and their potential biological significance.

## Introduction

Gene co-expression networks inform our understanding of cell and organismal function by encoding associations between genes. Associations between expression levels can indicate common function, and the number of connections can point to central or regulatory genes [1]. Due to the large dimensionality of gene expression data, often composed of several thousands of gene expression measures, a major tool in the analysis of co-expression is gene clustering: separating the genes into related groups, which can then be explored separately [2]. This drastically reduces the number of genes we need to consider at the same time and allows for the identification of hubs or centrally connected genes that can be used to inform further experimental validation [3,4].

The question is, given a co-expression network, how should we cluster the genes? The general idea behind several methods is to look for similar genes, as these are expected to be involved in related biological functions. However, several definitions of similarity have been used. The most basic measure of similarity borrows from classical morphological integration theory and attempts to find gene modules based on their correlations. In this context, genes in the same module are expected to be highly correlated and perform similar functions, while genes in different modules are expected to have low correlations [5–7]. Here, we refer to this classic pattern of higher within- than between-group correlation as assortativity, and to the groups as assortative modules. Other methods use the correlations to create other measures of similarity, which are then used as input to clustering algorithms. Weighted gene co-expression network analysis [WGCNA, 3] uses a power transformation of the correlation between gene expression (or a topological similarity measure built with these transformed correlations) [8,9] as a similarity measure that is then separated into assortative modules using hierarchical clustering. One of the main objectives of WGCNA is finding hub genes, which have high connectivity within modules and are clearly identified by hierarchical clustering. Other methods borrow from network analysis and attempt to explicitly maximize the Newman Modularity [10] of the weighted gene network. For example, Modulated Modularity Clustering [MMC, 11] uses an adaptive algorithm to find a non-linear distance between genes based on their correlations that maximizes the number of modules uncovered by maximizing modularity. Although these methods differ in their definition of similarity, they all impose an assortative structure on the gene expression network, in which similar genes are expected to be more correlated with each other than with other genes.

Clustering genes in tightly correlated modules aligns with the intuition that groups of genes performing similar functions should be highly correlated. However, imposing assortativity will necessarily ignore alternative organizations, if they exist, and could prevent us from fully understanding how transcriptional networks are organized. For example, Betzel et al. [12]

provides several examples of network organization besides assortativity, like non-assortative networks, in which vertices have more edges to vertices between communities than within; or core-periphery, in which a central community is connected to other communities, but the peripheral communities are not internally connected (Fig 1). These alternative architectures can also occur in the same network simultaneously [12,13]. We currently lack the empirical information on how common these patterns are in gene co-expression data, simply because our widely applied methods are completely blind to them due to their exclusive focus on assortativity. Given that other biological networks, like neuronal networks [12,14], exhibit clear evidence for these alternative organizations, there is no reason to think that a system as complex and as high-dimensional as the transcriptome should be limited to a single pattern of organization. To explicitly address this possibility of alternative organizations, we use a more general measure of similarity that allows us to find meaningful gene groups that are not necessarily assortative but still have clear biological interpretation. This approach is implemented in the weighted nested Degree Corrected Stochastic Block Model [14, wnDC-SBM, or SBM for brevity, 15], which has shown promising results in similar applications [16,17]. The SBM is different from other clustering methods in that it does not attempt to find assortative modules (i.e., modules with higher within- than between-module correlation). Instead, any information contained in the gene co-expression network can potentially be used to inform the clustering. Genes can be clustered together because they share similar patterns of connectivity to other genes, regardless of whether they are more strongly correlated with each other or with genes in other clusters. The word *information* here should be understood in the information theoretical

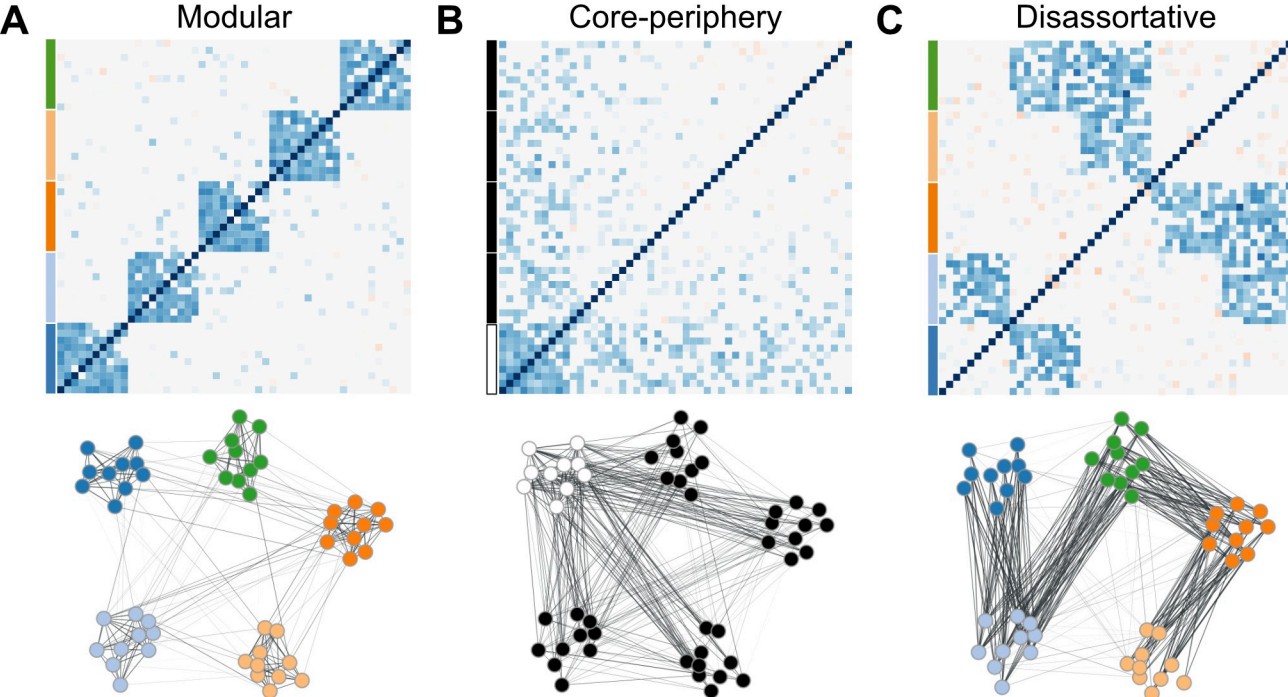

**Fig 1. Schematic representation of three network architectures.** Each panel shows the adjacency matrix (top) and the corresponding network diagram (bottom). A. Modular architecture: The network is composed of five distinct modules, each containing ten interconnected traits. Modules are connected by a few inter-module links. B. Core-periphery architecture: The network consists of a single densely connected core module with ten traits and a peripheral group with 40 traits. The peripheral group is connected to the core module, but has few internal connections. C. Disassortative architecture: The network comprises five groups, each with ten traits. Traits within each group are not interconnected but are instead connected to traits in other groups, forming a pattern of between-group connections.

sense: in the SBM, clusters are inferred so as to minimize the number of bits required to represent the network given the information about the clusters [18]. To be sure, the SBM can capture an assortative modular pattern if it is present, but it is general enough to also capture other network organizations [12,19]. Furthermore, even if, in the context of the SBM, assortativity is not the main driver of gene partitioning, it can still be used to interpret the clusters we obtain. By measuring the modularity of the identified clusters, we can compare networks with respect to their modularity without the problem of comparing a measure that was maximized to find the clusters in the first place. This opens the possibility of an unbiased comparison of the degree of modularity in different transcriptional networks (e.g., different cell types, tissues, species, etc.), which is a question that remains unexplored so far.

Here, using a multi-tissue RNAseq dataset from *Drosophila melanogaster*, we show first, that the SBM, a model with no free parameters, can find many more gene clusters than competing methods. Second, that such gene clusters are biologically meaningful, as revealed by highly specific gene ontology enrichment. Third, that biological meaning is not restricted to assortative modules as traditionally thought but extends to the non-assortative parts of the transcriptome. Our results highlight the importance of using clustering algorithms that don't rely on assortativity metrics to explore the structure of transcriptomes in a comprehensive and unbiased manner.

## Methods

### Gene expression measures

Elsewhere [20], we quantified whole-genome gene expression in hundreds of outbred *Drosophila melanogaster* female flies using a high-throughput RNAseq library preparation protocol [TM3seq, 21]. To build the gene co-expression networks, here we use a subset of the full dataset that includes: samples for two tissues, head (n = 212) and body (n = 252), individuals with the best coverage (average gene counts: head = 4.65M, body = 4.58M), and genes with moderate to high expression (average CPM > 5 and detected in every sample, head n = 5584 genes, body n = 5533 genes). The expression matrices used to generate co-expression networks correspond to VOOM-transformed gene counts (S1 Data) [22] where the effect of known and unknown [23] covariates was removed using the function removeBatchEffect from the R package limma [22]. Details on the collection of RNAseq data, library preparation, and processing of raw RNAseq counts can be found in [20].

### Gene co-expression network

Using the gene expression measures for both tissues, we generate co-expression network graphs. In theory, we could proceed using a full network in which all pairs of genes are connected but fitting the SBM with this fully connected graph is computationally too onerous. Moreover, given the very large number of correlations being estimated, many of these estimates are likely to be spurious or noisy. To mitigate these issues, we reduce the connectivity of the network by imposing a stringent Benjamini-Hochberg (BH) false discovery rate (FDR) cut-off on the edges. This approach removes edges with a large p-value associated with the correlation between the corresponding genes, thus retaining only the most statistically significant correlations that we are more confident reflect true biological relationships. As edges are removed, some genes with only non-significant correlations become disconnected from the rest of the network and can be removed. By gradually reducing the FDR threshold, we reduce the density of the gene network while attempting to keep as many genes as possible, until we arrive at a viable network with which to fit the SBM. We chose an FDR such that we reduce the graph density as much as possible (targeting a density of around 5%, corresponding to about

500,000 edges in the graphs) while still retaining most genes. Using this heuristic, we arrive at an FDR of 1% for the head and 0.1% for the body datasets which kept most of the genes (94.2% in the head:5261, and 92.6% in the body:5124) while reducing the graph density to a manageable level for use in the SBM, allowing the SBM to be fit in under a week of computational time. This same set of genes is used to compare two clustering methods: WGCNA and SBM. This choice of FDR is arbitrary but represents a pragmatic trade-off between computational feasibility, network sparsity, and retention of biologically relevant information. The chosen thresholds aim to strike a balance between these competing factors and provide a reasonable starting point for the application of the SBM to these large gene co-expression networks. Alternative thresholds could have been chosen, potentially leading to different downstream results, and to assess the effects of the FDR choice, we performed a sequence of SBM fits with a random subset of around 200 genes, showing that the main results should be robust to wide range of FDR choices (S1 Fig).

## Edge weights

Each method uses different edge weights for the network graph. WGCNA can use the fully connected graph, so we maintain all edges in this method. We use the topological overlap matrix (TOM) similarity in WGCNA. We use the low-density graph described above for the SBM, with the edge weight given by twice the inverse hyperbolic tangent transformed Spearman correlations between gene expressions. This transformation allows the edge weights to be modeled by normal distributions in the SBM, as we discuss below.

## Stochastic Block Model

The Weighted Nested Degree Corrected Stochastic Block Model [24] is a Bayesian generative model that attempts to find the partition with the highest posterior probability given the observed network and edge weights. Broadly speaking, this is achieved by dividing the network into groups of genes, called blocks, and modeling the weight and existence of a link between two genes in a network solely on their belonging to a particular block. So, genes with similar patterns of connections tend to be clustered in the same block. The degree correction refers to a modification of the standard Stochastic Block Model that allows genes with different degrees to be clustered in the same block [see 15 for details].

If $b$ is a particular partition of the genes in the weighted gene network $A$, we write a model that generates $A$ with probability given by $P(A|b,\theta)$, where $\theta$ stands in for any extra parameter we need besides the group partition $b$. With this model, we can write the posterior probability of the block partition $b$ given the observed network:

$$P(b \mid A) = \frac{P(A \mid \theta, b)P(\theta, b)}{P(A)}$$

where $P(A)$ is a normalization constant. In the canonical form of the Stochastic Block Model, the parameters $\theta$ are related to the expected number of edges between and within blocks. Specifically, the model would include parameters related to the expected number of edges between blocks $r$ and $s$, and parameters representing the expected degree of nodes in block $r$. These parameters are used to generate edges independently according to a Poisson distribution, meaning that the actual number of edges between blocks and the degrees of nodes can vary from the expected values. However, here, we use the microcanonical formulation from [14]. The microcanonical formulation of the Stochastic Block Model does not have any free parameters $\theta$. Instead, it imposes hard constraints on the network structure, requiring that the networks that are given non-null probabilities by $P(A|b)$ have exactly the same number of edges

between blocks and the same degree sequence as the observed network. In this formulation, the total number of edges between blocks $r$ and $s$ is fixed to the observed value $e_{rs}$, and the degree of each node is also fixed to its observed value $k_i$. This means that the microcanonical SBM generates networks with exactly the same block-wise edge counts and degree sequence as the observed network. Furthermore, the microcanonical formulation allows for the exact calculation of the probability of a network given a partition, which is useful for model selection and hypothesis testing. The probability of a network given a partition under the microcanonical assumption is simply the number of ways of observing that network divided by all possible networks that respect the constraints imposed by the block structure and the degree sequence. The mathematical expression for this probability is not particularly illuminating, so we refer the reader to [15] for the gory details. To provide some intuition about the idea behind this model, we can think about how, given the community labels, we can represent the network in a compact way using only the total number of edges (and their weights, see below) between and within each block. The inference task is therefore to find the partition $b$ that has maximal probability given the observation of the measured network, which is determined by the the constraints imposed by the edges, weights, and degree sequence.

**Description length.** We can also describe this inference of the optimal partition in the context of information theory. The posterior probability of the block partition can be written as:

$$P(b \mid A) \propto \exp(-\Sigma)$$

Where $\Sigma = -\log[P(A \mid \theta, b)] - \log[P(\theta, b)]$ is called the description length of the gene network $A$, and has an information-theoretic interpretation, being the amount of information required to encode the network given $\theta$ and $b$. So, finding the partition with maximal posterior probability is the same as minimizing the description length, or, in other words, the chosen partition $b$ is the one that allows us to describe the network using the least information, i.e., the least number of bits.

The two terms in $\Sigma$ also allow us to understand why this method offers intrinsic protection against overfitting. The first term $\log[P(A|\theta,b)]$ corresponds to the log-likelihood of the observed network. Increasing the number of blocks allows this likelihood to increase as the degrees of freedom of the model increase. But, the second term, $\log[P(\theta, b)]$ functions as a penalty that increases for complex models with many blocks, and the description length cannot decrease for overly complex models that have more blocks than warranted by the data. So, the selected partition with the minimum description length will necessarily be the simplest partition with similar explanatory power, avoiding overfitting and fully using the available statistical evidence. For example, the SBM would not detect modules that appear in random networks due to statistical fluctuations, in contrast to modularity maximization, which finds spurious modules in random networks [18,19,25]. We can also use the description length as a principled method for comparing models that simultaneously considers fit to data and model complexity.

**Weighted SBM.** The weights on the edges can be modeled in the SBM using different distributions depending on the edge weights. When edge weights are correlations, which are continuous numbers that vary between -1 and 1, it is natural to use some transformation to map the correlations onto the real numbers. The transformation we use is twice the arctanh transformed correlations as the edge weights and model these weights using normal distributions. In the SBM, the weights are modeled in much the same way as the links between networks, in that the mean and the variance of the observed edge weights between two blocks are a function only of the block structure, i.e., genes in the same block have a similar probability of being

connected to other genes and the value of the weights in these edges comes from the same normal distribution.

**Nested SBM.**    The nested SBM uses a series of hierarchical priors that greatly increase the resolution of detected blocks and allow the model to be fully non-parametric. This nested structure allows for the identification of more and smaller blocks that are statistically supported than other clustering methods [15]. This is achieved by treating the gene block partition as the nodes in a nested series of networks, which are then clustered using the same method. So, the genes are clustered in blocks, and these blocks are also clustered in higher-level blocks, and so on, as required to minimize the description length of the gene network (see diagram in Fig 2). The model estimates the number of levels in the hierarchy and the number of blocks in each level. Since the model is generative, we can use posterior samples of the partitions to quantify the uncertainty in any quantity estimated by the model, like the number of levels in the hierarchy, or the number of blocks at each level.

**Fitting the SBM.**    For details on the implementation of the SBM, see [15] and [14]. All SBM were fitted using the graph-tool python library [26]. The fitting process consisted of three steps. First, an initial partition of genes into blocks at each level of the SBM hierarchy was obtained using the *NestedBlockState* function. Next, the block partition was refined using the *mcmc_anneal* function, which uses Markov Chain Monte Carlo (MCMC, [15]) and simulated annealing [27] to find a better network partition (i.e., one with larger posterior probability and smaller description length). Annealing is not always necessary, but in our data it proved to be efficient in reducing computational time. Next, the *mcmc_equilibrate* function was employed to find a partition where subsequent proposals did not improve the posterior probability of the current partition for at least 1000 proposals. At this stage, the block partition was considered equilibrated, allowing for posterior sampling using MCMC. Finally, the posterior sampling was conducted for 1000 iterations using the *mcmc_equilibrate* function, and the median partition of this posterior sample was used for subsequent analysis.

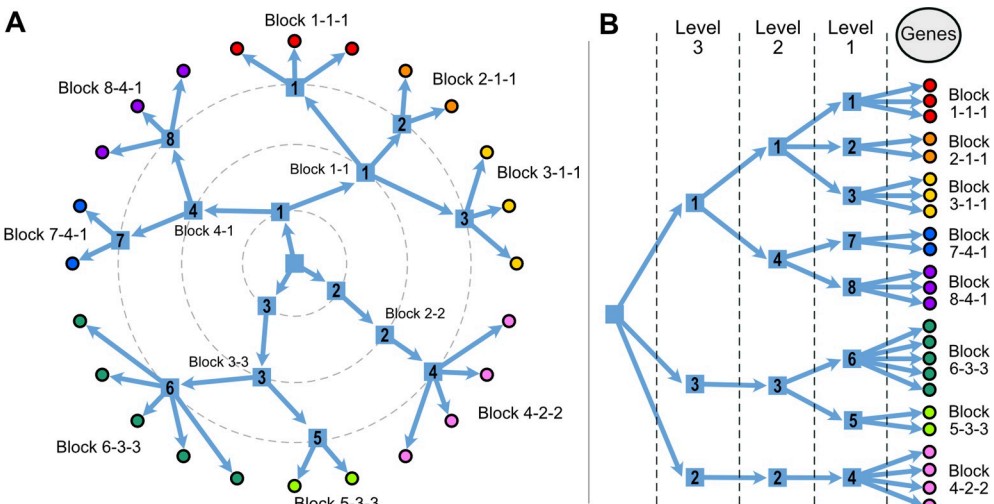

**Fig 2. Schematic representation of the clustering in the SBM.** Genes are clustered into level-1 blocks, level-1 blocks are clustered into level-2 blocks, and so on. A. Circular representation of the clustering we use in the following figures. Block names are constructed by following the hierarchy, starting at level 1. So in this example, the level-1 block 8 can also be referred to as 8-4-1. B. A tree-like representation that highlights the hierarchy in the nested SBM. Each level-2 block is composed of all the genes in its child level-1 blocks, each level-3 block is composed of all the genes in its child level-2 blocks, and so on.

**Modularity and Assortativity.**   Although the nested SBM does not attempt to find the partition of genes that maximizes modularity (see definition below), when using this method we can ask if the inferred partition is modular or not by calculating the Newman modularity at each level of the hierarchy. Newman Modularity is calculated at each nested level using:

$$Q = \frac{1}{2E} \sum_r e_{rr} - \frac{e_r^2}{2E}$$

where $e_{rr}$ is, by convention, twice the sum of edge weights internal to group $r$, $e_{rs}$ is the sum of edge weights between groups $r$ and $s$, $e_r = \sum_s e_{rs}$, and $E$ is the sum of all weights. Newman modularity quantifies the intuition that genes in the same module should be more connected than across modules by comparing the within-group connections ($e_{rr}$) to the expected value of the connections across all the groups $\left(\frac{e_r^2}{2E}\right)$. The higher the difference between correlations within- and between-groups, the higher the value of $Q$.

We follow [14] and further decompose the contribution of each Level-1 block to the modularity by defining the local assortativity of a block as:

$$q_r = \frac{B}{2E} \left( e_{rr} - \frac{e_r^2}{2E} \right)$$

where $B$ is the number of blocks. Using this definition, modularity is just the average local assortativity, $Q = \frac{1}{B} \sum_r q_r$. Blocks with a positive value of $q_r$ show assortativity, while blocks with negative values of $q_r$ show disassortativity, with more connections across other blocks than within. We note that $q_r$ values are not independent, and changes in the assortativity of one block can lead to changes in assortativity for the other blocks. Furthermore, the possible values of modularity for a particular network depend on other aspects of the network other than the community structure, like degree distribution, and therefore comparisons of modularity values across different networks should be made with caution [28]. Even so, if we consider that the modularity value $Q$ is meaningfully estimated, the $q_r$ values can be interpreted as the contribution of each block to the total modularity.

## WGCNA

We use WGCNA to cluster the genes into modules using the topological overlap measure (TOM) similarity with a soft threshold of 6 in a signed similarity measure. WGCNA produces modules by cutting the hierarchical clustering tree at different heights, and we use the dynamic cutting option to create the modules. We use a signed network (as opposed to ignoring the sign of the correlation between genes) because inspection of the gene network graph reveals large groups of genes linked by negative correlations in our data, suggesting a large-scale structure that would be obscured by using the unsigned method. Signed similarity has been shown to lead to more robust modules [29], and in tuning WGCNA we were able to cluster more genes and find more modules using the signed method. The choice of using WGCNA with the full dense network and not the FDR trimmed is intentional, as we feel it is important to compare the methods using their recommended workflows. The use of p-value FDR trimming is not a part of the standard WGCNA method and would be a novel and untested modification. Additionally, WGCNA already has a strategy to handle edges with small correlations through its use of soft-thresholding. The way we process the gene expression data is an integral part of the clustering method we are comparing, even if they lead to different networks. Our intention here was to compare the full workflow, not the specific clustering on a particular network. Given this, we present the results using WGCNA's proposed workflow below and add the FDR

trimmed version of WGCNA to the S2 Fig. Using the trimmed version of the network in WGCNA does not significantly change the main results.

## Simulations

Given that there are several articles using simulations to compare the clustering obtained via SBM to other modularity maximization methods, we do not employ extensive simulations here. For example, see [18] and references therein for an overview of the performance of SBM in the context of inferring community structure, and see [19] for a version of SBM specifically developed to overcome the problems of modularity maximization when searching for assortative communities. Instead, we use a simple simulation to illustrate our network building and community detection workflow and to show the different aspects of network architecture that can be captured by both methods. We start with a known modular correlation matrix (S2 Data), with 50 traits grouped into 5 assortative modules. These five modules are further grouped such that there are 2 higher level groupings with two modules each, and the fifth module is equally correlated with the other 4 (Fig 3A). Using this correlation matrix, we sample 1000 draws from a multivariate normal distribution, generating observations that follow the known correlation pattern (S2 Data). We then measure the Spearman correlation across these simulated observations and produce an FDR trimmed weighted network (Fig 3B) that is then clustered using SBM (Fig 3C) and WGCNA (Fig 3D). Next, we modify the original correlation matrix to include a non-assortative module. We accomplish this by taking the first 5 traits and setting the correlations between them to a value lower than the across module correlations (Fig 3E). We then repeat the sampling and inference procedure, producing a new trimmed network (Fig 3F), an SBM fit on this network (Fig 3G), and a WGCNA fit on this network (Fig 3H).

## Gene Ontology enrichment

We assess the biological relevance of the clustering obtained by each method by comparing their gene ontology (GO) enrichment. We filter GO enrichment p-values using a BH FDR rate of 5%, with a minimum of 4 genes in each enriched set. All gene ontology analyses were done using the clusterProfiler R package v4.2.2 [30] and the Org.Dm.eg.db database package v3.15 [31].

## Results

### Simulations

Our simulations illustrate the differences between the SBM and WGCNA in capturing various aspects of network architecture. When applied to the simulated data with assortative modular structure (Fig 3A), both methods were able to recover the underlying community structure to a certain extent. The SBM (Fig 3C) correctly identified the five modules and further grouped them into two higher-level clusters. WGCNA (Fig 3D) detected 4 modules, failing to separate the first two modules. Furthermore, when a non-assortative group was introduced into the correlation matrix (Fig 3E), the two methods exhibited markedly different behaviors with respect to this non-assortative group. The SBM (Fig 3G) was able to identify the non-assortative cluster (highlighted in red) and correctly assigned its traits to a separate block, showing its ability to detect various types of network structures beyond assortative communities. In contrast, WGCNA (Fig 3H) failed to recognize the non-assortative module and instead grouped its traits with one of the assortative modules (colored in teal), highlighting its limitation in handling non-assortative network architectures. Indeed, there is no way for the hierarchical tree to identify this group as a separate cluster. These results underscore the versatility of the SBM in

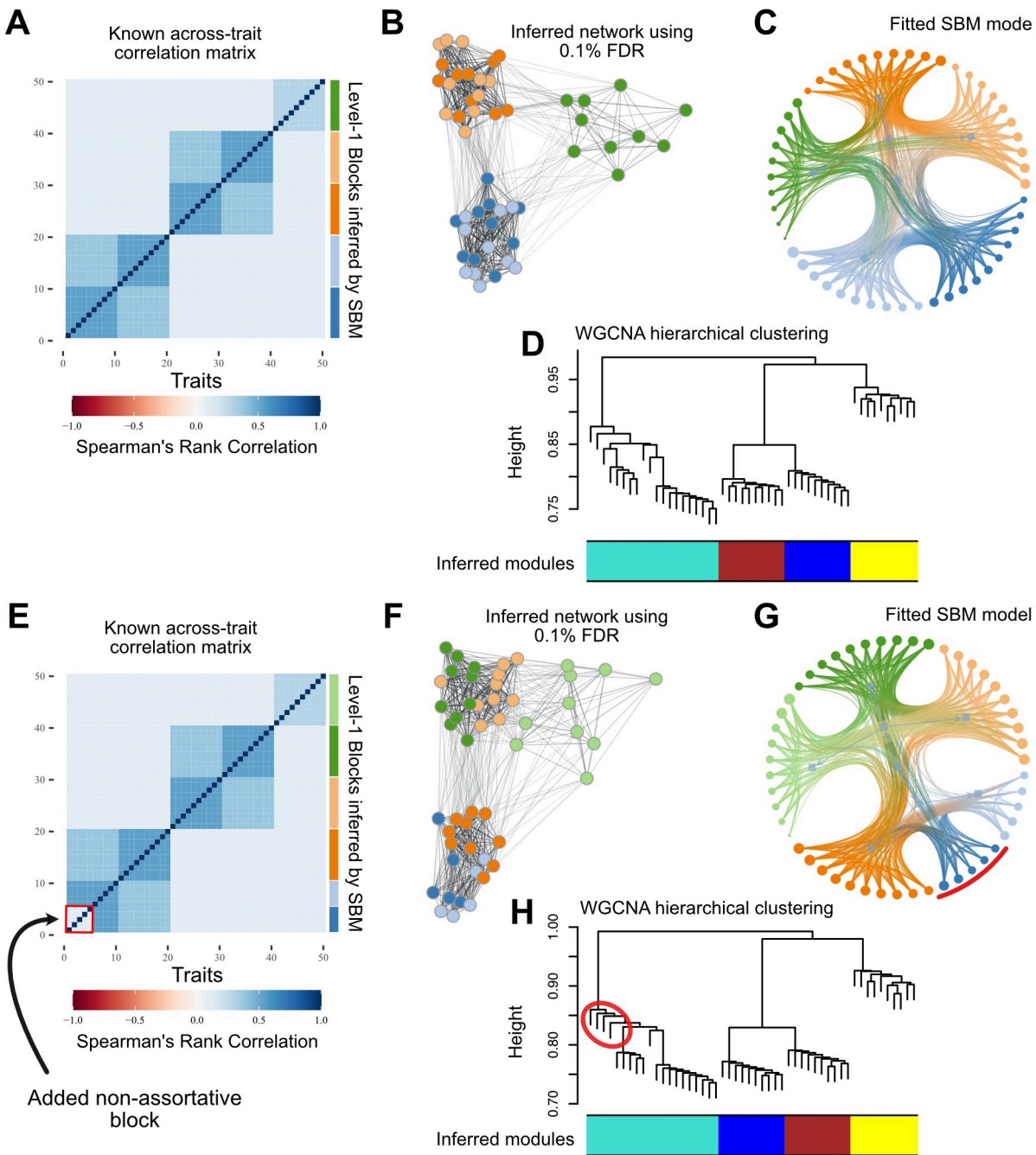

**Fig 3. Simulations comparing the Stochastic Block Model (SBM) and Weighted Gene Co-expression Network Analysis (WGCNA) in detecting assortative and non-assortative network structure.** A. Known modular correlation matrix with 50 traits grouped into 5 assortative modules, further organized into 2 higher-level groupings and a fifth module equally correlated with the others. B. FDR-trimmed weighted network generated from observations sampled from the correlation matrix in (A). C. SBM fit on the network in (B), correctly identifying the five modules and their higher-level organization. Edges are colored according to the block assignment of the vertices. D. WGCNA fit on the network in (B), detecting the only 4 assortative modules but not the higher-level groupings. E. Modified correlation matrix with a non-assortative module (first 5 traits, highlighted in red) introduced. F. FDR-trimmed weighted network generated from observations sampled from the correlation matrix in (E). G. SBM fit on the network in (F), correctly identifying the non-assortative module (in blue with red arc) and the assortative modules. H. WGCNA fit on the network in (F), failing to recognize the non-assortative module and grouping its traits (circled in red) with an assortative module (teal module). These simulations show the ability of SBM to capture both assortative and non-assortative network structures, as well as hierarchical organization, compared to WGCNA, which is primarily designed for detecting assortative communities.

capturing diverse network architectures, including both assortative and non-assortative modules, as well as hierarchical organization.

The simulations also illustrate the effectiveness of the FDR-based network trimming approach in reducing network density while preserving biologically relevant connections. The trimmed networks (Fig 3 panels B and F) maintained the essential structure of the original correlation matrices while removing potentially spurious edges that we lack the statistical power to identify.

## Gene clustering

To assess the consequences of assuming that communities in transcriptional networks are assortative, we compared the performance of a clustering algorithm that relies on assortativity (WGCNA) to the performance of the SBM. For this, we run both clustering algorithms on the same gene co-expression matrices. To distinguish between gene clusters derived from the SBM and WGCNA, we refer to the former as 'blocks', and to the latter as 'modules'. Gene clustering for all methods is presented in S1 Table.

Using the SBM, in both *Drosophila* head and body RNAseq datasets, we were able to cluster all genes, identifying a nested partition with 5 levels (Fig 4). We obtain 2 blocks for both tissues at level 5 (the coarsest); 3 blocks for both tissues at level 4; 6 block for the head and 9 blocks for the body in level 3; 21 blocks for both tissues at level 2; and, finally, 82 blocks for the head and 78 blocks for the body at level 1. In what follows, when discussing specific SBM blocks, we either explicitly define which level of the nested hierarchy we are referring to or give the full path to a given block. For example, level-1 block 12 can also be referred to as 12-7-2-2-1 (see Fig 2 for an illustration on how to interpret these labels).

In contrast with the SBM, WGCNA was able to cluster only 30–40% of the genes. These 2118 genes in the body and 1600 genes in the head were partitioned into 7 modules in both tissues. To assess whether the gene clusters inferred by each algorithm are similar, we compared the results of WGCNA to the SBM blocks at level 3 (Fig 5). We focused on level 3 instead of level 1 (the finest level) because the number of blocks at this level (6 in the head and 9 in the body) are similar to the number of modules in WGCNA (7 modules for both tissues). Overall, the partitions are different, but WGCNA and the SBM do capture some common signals, evidenced by the tendency of Level-3 blocks that share the same Level-4 blocks to be grouped into the same modules in WGCNA. For example, Level-3 blocks 0, 2, 5, and 6 in the body are split between modules 3 and 4, and these blocks are all in the same Level-4 block 0, suggesting some similarity that could explain the WGCNA clustering. Blocks 7 and 9 are both fully assigned to module 2. Also in the body, we find a similar pattern for Level-3 blocks 1, 3, and 4, which are mostly split between modules 1 and 2. In the head, Level-3 block 4 is all assigned to modules 1 and 3. Level-3 blocks 1 and 2 are mostly split between modules 1 and 3, and both are in Level-4 block 2. Importantly, level 3 is an intermediate level in the clustering hierarchy resolved by the SBM, and at finer levels (i.e., level 2 and 1) the gene groups are smaller and functionally more specific.

## Modularity and assortativity

Because the SBM does not use modularity maximization to find communities, we were able to use the resulting clustering to measure, in an unbiased manner, the assortativity of individual blocks and the overall degree of modularity of the transcriptional networks in the head and the body. We find that modularity and assortativity are markedly lower in the body (Fig 6). Several blocks in the body have negative assortativity (being more connected across blocks than within), and the maximum value of modularity is 0.035 at level 4 of the nested hierarchy. Even

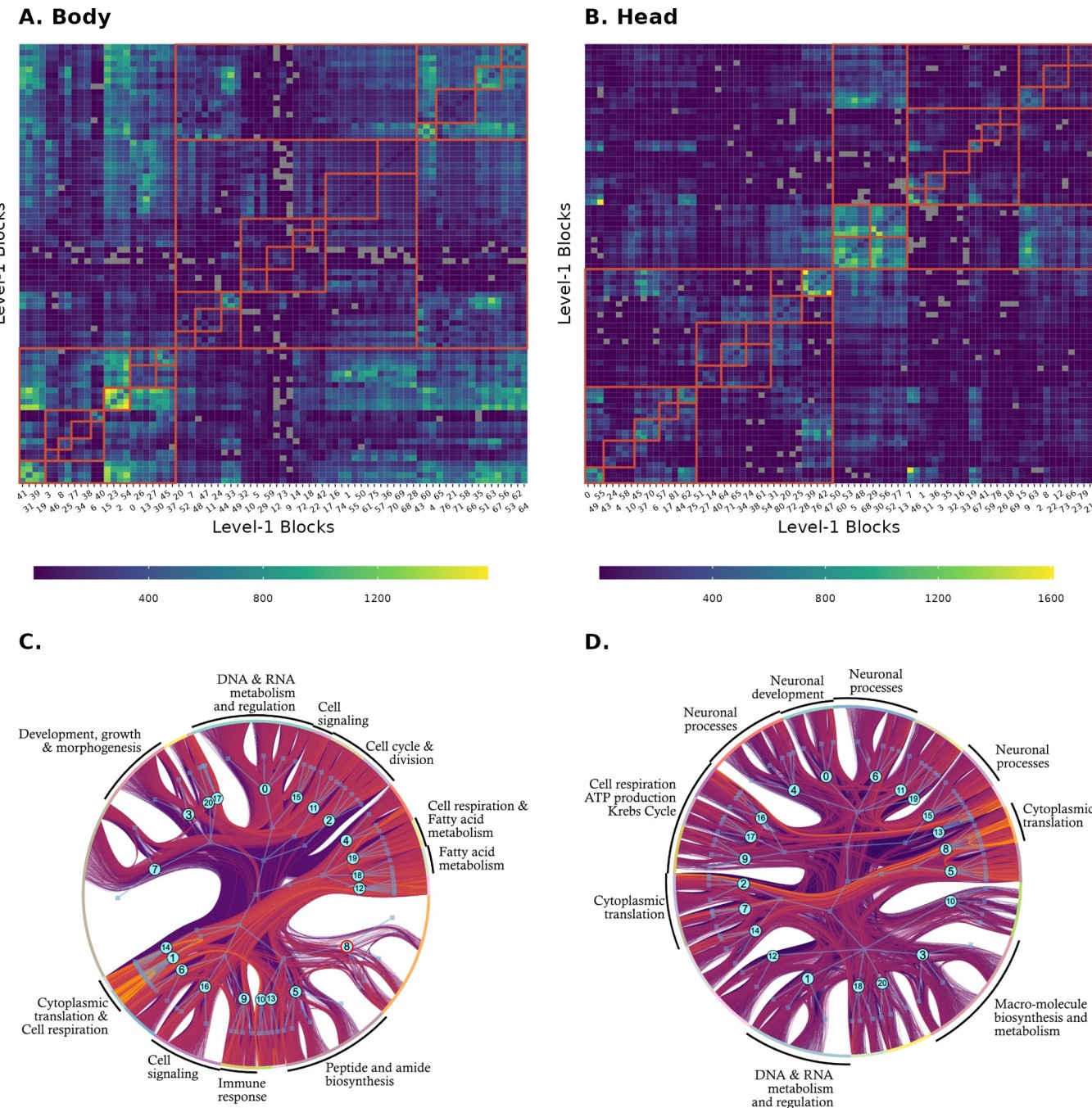

**Fig 4. Matrix and graph representations of the SBM clustering.** A and B: SBM Level-1 blocks are colored by the number of edges within and between blocks. Gray squares represent pairs of unconnected blocks. The upper levels of the nested hierarchy are shown by the red lines. C and D: A full representation of the fitted block model. Genes are shown at the perimeter, colored by their level 2 blocks. The internal graph shows the hierarchical structure of the fitted SBM. Numbers in blue circles correspond to the level-2 block. Arrows between level-1 blocks and genes are omitted, unlike Fig 1. A subsample of 30.000 edges is shown connecting the genes, and edges are colored according to their transformed weights, with more positive weights plotted on top and more yellow. External labels refer to a non-exhaustive subset of level-2 blocks with clear biological functions inferred from interpreting GO enrichment. Level-2 block 8 in the body, with the blue circle highlighted in red, is the only level-2 block with no GO enrichment.

so, several blocks show GO enrichment throughout the distribution of assortativity. In the head, overall modularity is higher, with a peak at 0.14 in level 3. This is still a relatively low

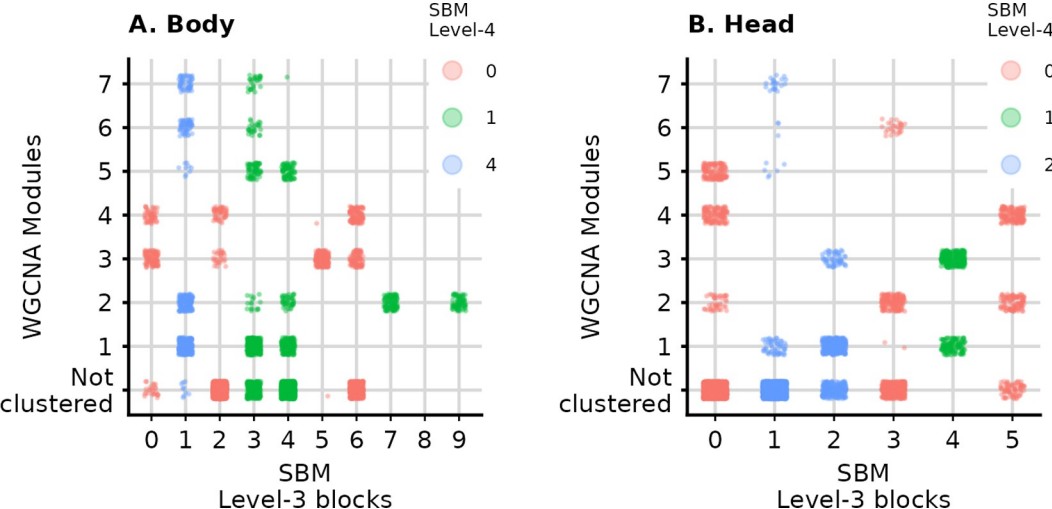

**Fig 5. Comparison of the clustering in WGCNA and levels 3 and 4 of the SBM hierarchy for the gene expressions in the body (left) and the head (right).** Each point corresponds to a gene. The x-axis corresponds to the Level-3 SBM blocks, and the y-axis the WGCNA modules. Colors correspond to the (coarser) level 4 of the SBM.

value and illustrates how assuming the gene network should be modular can prevent us from finding an informative clustering. All but 5 blocks in the head show positive assortativity, and again GO enrichment is present throughout the assortativity range (Fig 6).

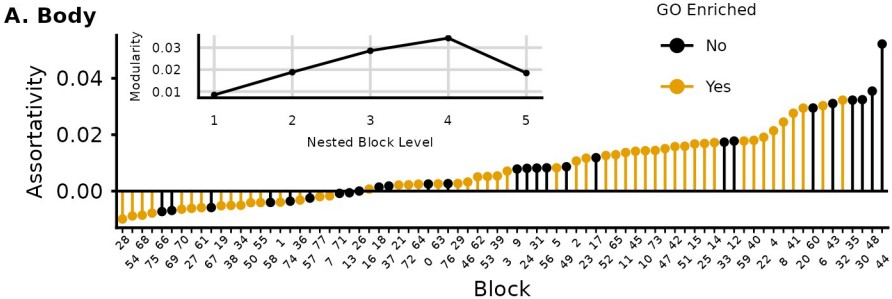

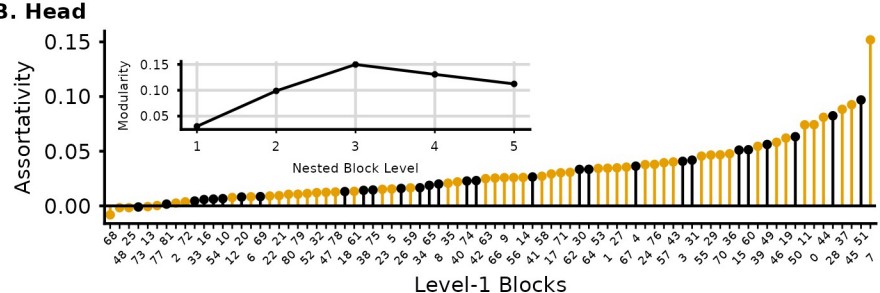

**Fig 6. Assortativity measured in the SBM level-1 blocks and Newman Modularity (average assortativity) at each level of the SBM hierarchy (inset).** GO enriched blocks are shown in yellow and appear throughout the distribution of assortativity. Modularity is much higher in the head, and it peaks at level 3, dropping in upper levels. Body has a much higher number of non-assortative blocks and lower modularity at all levels. Modularity peaks at level 4 in the body and drops strongly at level 5. Interestingly, the 4 most assortative blocks in the body do not show significant GO enrichment.

## Gene Ontology enrichment

Most blocks in SBM show significant GO enrichment (Table 1). Enriched level-1 blocks show between 1 and 202 enriched terms, with a mean of 24 terms and median of 13 terms. Level-2 blocks show between 2 and 297 enriched terms, with a mean of 58 terms and median of 38 terms. Furthermore, several blocks show remarkable consistency in their enrichment. For example, Level-3 block 0 in the head is related to neural signaling, sensory perception, and signal transduction. Examining lower levels of the hierarchy, we see that often the daughter blocks at Level-2 are also enriched with generally similar terms, as expected, but these tend to become more specific as we go down the hierarchy. For example, Level-2 blocks 4 and 6: (4-0-0-0) G protein-coupled receptor signaling pathway, detection of light stimulus, phototransduction; (6-0-0-0) synapse organization, axon development, cell-cell signaling, behavior. Many of these enrichments are exclusive to one of the level-1 blocks. Most other Level-2 and Level-1 blocks are readily identifiable as related to development, DNA transcription, cell respiration, cell cycle regulation, immune response, sugar metabolism, among others (Fig 4). All WGCNA modules show GO enrichment (but modules 5, 6, and 7 in the body show only one or two enriched terms, and could be false positives. The more convincing specific enrichments show several related enriched terms). The remaining modules show between 20 and 462 enriched terms, with a mean of 116 terms and median of 58 terms. In general, these enrichments tend to be less specific than the SBM blocks, spanning several biological processes. S2 Table shows GO enrichment for all SBM blocks and WGCNA modules.

**Notable individual clusters.** Level 2 block 0-0-0 in the head is one of the easiest to interpret, being entirely related to nervous tissue function. S3 Fig shows the top 8 GO categories for each of the level-1 blocks in block 0-0-0, and the most neuronal enriched WGCNA grouping, module 4. The SBM blocks separate vesicle exocytosis, neuronal differentiation, phototransduction, synaptic signaling, and, interestingly, there is a block related to mRNA processing, which is notable given that alternative splicing is thought to be more common in brain tissues [32]. WGCNA module 4 recovers some of this enrichment but in a less granular way. The cell adhesion and developmental part of the enrichment in block 0-0-0 is separated between WGCNA modules 4 and 5. Some of the level-1 blocks shown in S3 Fig are among the most assortative (above 0.03, see Fig 6B), and so are prime candidates for detection in WGCNA. The alternative splicing module has a much lower assortativity, so it is not surprising that WGCNA could not detect it.

Some of the most specific enrichments in the SBM are the translation-related blocks. In both body and head, ribosomal proteins are clustered in small and highly enriched level-1 blocks: 6 level-1 blocks in the head and 11 in the body are composed of virtually only ribosome-related protein genes. All are very small, being composed of between 10–30 genes, have low assortativity (Fig 7), and are enriched for very few terms, almost all related to translation. In the body, all of these translation blocks are grouped in level-4 block 1; in the head, they are split between level-4 blocks 1 and 2. Both groups are visible in Fig 4. There is no equivalent module in WGCNA, but all translation-related genes are in the same much larger modules (module 2 in the head, 295 genes; and module 2 in the body, 345 genes), both of which show

**Table 1. Fraction of blocks at each level of the SBM hierarchy that show significant GO enrichment at the 5% FDR level with a minimum of 4 genes in the enriched set.**

| Tissue | Level 1 | Level 2 | Level 3 | Level 4 | Level 5 |
|---|---|---|---|---|---|
| Head | 65% (53/82) | 100% (21/21) | 100% (6/6) | 100% (3/3) | 100% (2/2) |
| Body | 65% (51/78) | 95% (20/21) | 100% (9/9) | 100% (3/3) | 100% (2/2) |

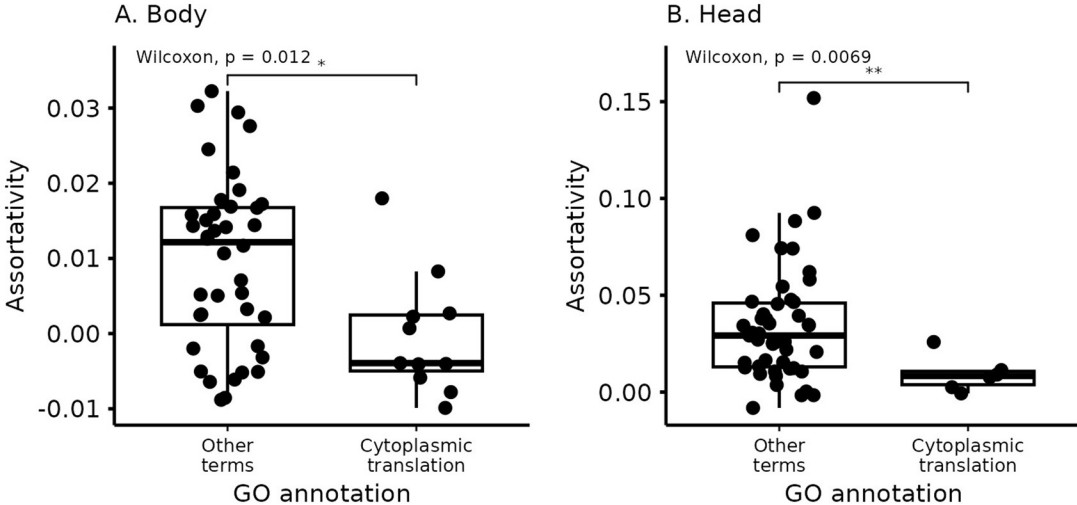

**Fig 7. Comparison of assortativity values between level-1 blocks enriched for cytoplasmic translation and all other blocks. Blocks enriched for cytoplasmic translation tend to be less assortative.**

enrichment for translation but also several other categories. In the body WGCNA module 2, we see 68 enriched terms related to translation, cell respiration, and several small molecules' metabolic processes; in module 2 of the head tissue, we see 35 enriched terms related to translation, cell respiration, and muscle development. The level-2 clustering of level-1 blocks in the SBM is also informative. In the head, all the translation level-1 blocks are in their own level-2 blocks (8-4-1-1, 7-2-2-1, and 2-2-2-1). In contrast, in the body, the level-1 translation blocks sometimes share level-2 blocks with cell respiration blocks: 1-7-1-1 is composed exclusively of level-1 blocks related to translation, but block 14-9-1-1 is split into translation and mitochondrial respiration level-1 blocks. WGCNA also places cell respiration-related genes in the body on the same module 2.

## Discussion

Here, we have used the Stochastic Block Model to explore the organization of gene co-expression networks in female *Drosophila melanogaster*. The SBM, in contrast with other gene clustering methods, clusters genes by finding groupings that capture as much information on the network of interactions as possible, and was able to (i) cluster all genes into blocks; (ii) identify blocks with both, high resolution (few genes per block) and high functional content (significant GO associations); and (iii) identify blocks that are assortative (higher within- than between-block correlation) as well as non-assortative. This last point exemplifies the novelty of the SBM approach. Using the SBM implies a shift on how we explore co-expression networks: instead of assuming the network is modular and clustering genes based on this assumption, we uncover clusters based on their information content and ask if the resulting groups are modular. Surprisingly, the answer is not always.

### Community detection methods

Commonly used clustering approaches, that explicitly search for assortative modules, carry important downsides. Methods that use modularity maximization (WGCNA [3], MCMC [11]) are subject to know statistical problems, surprisingly being prone to both overfitting (finding modular community structure where there is none, [25]) and under-fitting (failing to

find modular structure), due to a problem known as the resolution limit, which causes small modules to be incorrectly clustered together in large networks [33]. Using WGCNA involves manually tuning several parameters: the choice of using a hard or soft threshold, the exponent in the threshold, the method of separating the genes included in the hierarchical clustering into the modules. These are free parameters that can drastically change the number of genes that are clustered and the number and size of modules. For tuning these parameters, the WGCNA workflow leans heavily on the expectation that gene co-expression networks should be approximately scale-free [9,34,35], but, despite its popularity, this expectation might be unwarranted [36–39]. Even with optimal parameters, WGCNA often fails to assign a substantial proportion of genes to any module. While WGCNA is rightfully popular and efficient in finding hub genes, if some functional gene group does not have a hub or has low average similarities, this group will never be identified. Both limitations potentially leave biological insight on the table by ignoring network structures that are different from what the method expects. In contrast, all the freedom in the SBM is restricted to the creation of the network, which we discuss below, and no assumption is made on the structure of the communities in the network. The clustering procedure is completely parameter free, and choices regarding how to model the weights between edges can be made by selecting the model with the shortest description length [15]. This is a significant advantage for applying the SBM in domains where we lack relevant domain expertise and can't easily tune the parameters for the other clustering methods. Furthermore, the SBM finds a much larger number of communities that are guaranteed to be statistically supported, greatly improving the resolution of the clustering and allowing for more precise biological interpretation of the resulting blocks.

One aspect we did not explore here is the estimation of the gene co-expression network itself, before any attempt at finding communities. Both methods used weighted networks: fully connected ones for the WGCNA (as per this methods' suggested workflow) and a sparser network for the SBM model fitting. Estimating these weights (gene expression correlations) is an error-prone process, as we are estimating many more weights than we have measured individuals, leading to potentially poor estimates [40]. WGCNA uses a soft-threshold approach to mitigate the impact of small correlations, where the correlation values are raised to a power (the soft-threshold parameter) to reduce the influence of weak correlations and emphasize strong ones. This soft-thresholding helps to alleviate the effect of noisy or spurious correlations in the network construction process. The FDR edge trimming approach used in this study is an attempt to maintain only edges for which we have sufficient evidence to include in the network. In contrast, the more widely used correlation value thresholding involves setting a fixed threshold for the absolute value of the correlation coefficient, below which edges are removed from the network. This approach can be problematic because it does not account for the statistical significance of the correlations and may lead to the removal of biologically relevant, but weak, edges. FDR trimming, on the other hand, provides a less binary way of deciding which edges to include in the network, balancing the need to control for false positives while still retaining edges that are likely to represent relevant associations, even if their correlation values are not particularly high. While the procedures we used here are commonplace, there are more principled ways of building co-expression networks [41], and this is an aspect of the usual transcriptomics workflow that could potentially see massive improvements in the near future. Methods like the graphical lasso have been used in this context [42–44], and the expectation is that, when compared to fully connected or thresholded networks, these inferred networks should provide much better estimates of gene-gene connections and weights. Additionally, it is possible to combine community detection via the SBM with network inference, simultaneously using information about community structure to inform the network inference and vice-versa [45].

## Modularity in gene co-expression networks

Beyond the methodological and practical advantages discussed above, the fact that the SBM does not find gene clusters by attempting to maximize their modularity has major implications for our understanding of biological networks, as it allows us to *measure* the modularity of a given network. In doing so, we find that *D. melanogaster* transcriptomes are organized into assortative as well as non-assortative gene clusters. The latter, however, could not have been identified by methods that assume assortative modules. The possibility of quantifying, in a continuous scale, the degree of assortativity of each gene block allowed us to compare the gene co-expression networks derived from head and body tissue, and uncover marked differences in their overall degree of modularity.

These results warrant a discussion about the origin of the assumption that gene expression networks are modular. Modularity, understood as the relative independence between groups of complex traits, is often invoked to explain the evolvability of complex phenotypes and has functioned as a unifying concept at several levels of organization with great success [7,46,47]. Traits in an organism need to have some level of integration, of interdependence, to form a functioning individual. This necessary interaction between parts poses a problem for understanding the evolution of complex traits, as interdependencies are expected to lead to important evolutionary restrictions [48]. Modularity provides a simple solution to this problem as it allows organisms to maintain their function unchanged by coordinating simultaneous evolutionary changes in all related traits while keeping unrelated traits undisturbed [49–52]. The conceptual usefulness of modularity has informed much of our thinking on how complex traits should be structured, producing a large body of literature dedicated to finding modules and testing for their existence [53]. A large part of the literature on modularity developed in the context of morphological traits, and morphological traits being organized into modules can be interpreted as a consequence of the very concrete structural and developmental constraints that lead to the formation and allow proper functioning of these individual body elements [54,55]. These constraints are easy to visualize, as morphological traits like bones and muscle have to fit together in order to function, and individuals in which perturbations are large enough to disrupt these couplings are not viable. The result is a modularity pattern that is kept stable by these structural and functional constraints [56–58]. However, no such clear structural and physical constraints exist on gene expression, and the interaction between groups of genes can happen through more dynamic and varied mechanisms. While we might expect related genes to be co-expressed and therefore highly correlated, non-linear phenomena can lead to a complete decoupling of the expression levels of co-expressed genes. For example, the effect of gene A on gene B could have a saturation point after which increasing expression of gene A no longer leads to higher levels of gene B, and no correlation is detected in this regime, even if the genes are co-expressed [59]. The marked difference in the level of modularity across the two tissues in our samples illustrates just how variable modularity can be, even within the same species, sex, and population. Furthermore, modularity is not a necessary feature of biological organization [even in the case of evolvability, see 60,61,62], and only searching for modularity can blind us to alternative organizations, as we have shown. Indeed, the profound interconnectedness of gene regulation networks has led to a small revolution in our understanding of disease and complex traits [63]. The very high dimensionality of gene co-expression networks also allows for genes to be similar in ways that do not lead to high correlations. For example, two genes might be connected to the same genes in different communities, but not among themselves (like the non-assortative block in Fig 3E). This similarity would likely be missed by modularity maximization or hierarchical clustering because these genes would not form a classic assortative unit. Meanwhile, the SBM would correctly identify these genes connecting two

modules as being similar due to their shared connectivity pattern. Having access to these types of blocks, which are real but non-assortative, could bring new insight into the organization of gene co-expression networks, illustrated by the relation we describe between non-assortativity and cytoplasmic translation.

## Conclusion

Here, we find that non-modular blocks are widespread in gene co-expression networks, and that the evidence for their functional relevance is as strong as for modular blocks. This highlights the need to incorporate other sources of information, beyond assortativity, when exploring biological networks. More studies using methods that don't rely on modularity maximization will be needed to determine whether there are general patterns of non-modular organization.

## Supporting information

**S1 Data. VOOM transformed, batch corrected, gene expression levels.**
(XLSX)

**S2 Data. Data related to the simulations in Fig 2.**
(XLSX)

**S1 Fig. Exploration of the effect of FDR choice in the SBM clustering.**
(PDF)

**S2 Fig. WGCNA clustering using FDR trimming.**
(PDF)

**S3 Fig. Enriched GO categories in a level-3 block in the head (0-0-0), related to neural signaling.**
(PDF)

**S1 Table. Gene clustering using SBM and WGCNA.**
(XLSX)

**S2 Table. GO enrichment for all methods and tissues.**
(XLSX)

**S3 Table. Summary statistics for all SBM blocks.**
(XLSX)

## Acknowledgments

We thank all members of the Ayroles lab for their support. We thank Monique Simon and Cara Weisman for their thoughtful comments on an earlier version of the manuscript. We thank Tiago Peixoto for help in using graph-tool. We also acknowledge that the work reported in this paper was substantially performed using the Princeton Research Computing resources at Princeton University which is a consortium of groups led by the Princeton Institute for Computational Science and Engineering (PICSciE) and Office of Information Technology's Research Computing.

## Author Contributions

**Conceptualization:** Diogo Melo, Luisa F. Pallares, Julien F. Ayroles.

**Data curation:** Luisa F. Pallares.

**Formal analysis:** Diogo Melo.

**Funding acquisition:** Diogo Melo, Luisa F. Pallares, Julien F. Ayroles.

**Investigation:** Diogo Melo, Luisa F. Pallares, Julien F. Ayroles.

**Methodology:** Diogo Melo, Luisa F. Pallares.

**Project administration:** Julien F. Ayroles.

**Resources:** Julien F. Ayroles.

**Software:** Diogo Melo, Luisa F. Pallares.

**Supervision:** Julien F. Ayroles.

**Validation:** Diogo Melo.

**Visualization:** Diogo Melo.

**Writing – original draft:** Diogo Melo.

**Writing – review & editing:** Diogo Melo, Luisa F. Pallares, Julien F. Ayroles.

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
