## [Decision Letter · Decision Letter 0]

9 Feb 2024

Dear Diego Melo et al.,

Thank you very much for submitting your manuscript "Reassessing the modularity of gene co-expression networks using the Stochastic Block Model" for consideration at PLOS Computational Biology.

As with all papers reviewed by the journal, your manuscript was reviewed by members of the editorial board and by three independent reviewers. In light of the reviews (below this email), we would like to invite the resubmission of a significantly-revised version that takes into account the reviewers' comments.

All reviewers agree that there is utility in having models that can infer modularity in a network without imposing it a priori (for reasons the authors explain). It seems to me and the reviewers that the weighted Stochastic Block Model (SBM) is capable of identifying more modular clusters than other methods that use assortativity a priori. The two main issues that prevented us from being fully convinced are that some of us: 1) had difficulty understanding parts of the model, and 2) could not see whether the comparisons between SBM and other models (mainly WGCNA) were appropriate for distinguishing SBM as the preferred model. I shall discuss these two issues in turn.

Description of the model both conceptually and mechanistically. Conceptually, I agree with Reviewer 1 that a better biological motivation is needed for introducing an assortativity-free model. We might easily imagine that some gene groups are not assortative, but real examples, and explanation or reasoning as to why they are not assortative, would go a long way to motivating the need for SBM (see Reviewer 1 paragraph on "A grounding in real biological examples"). Mechanistically, Reviewer 1 and 2 had some uncertainty about the model and its implementation, which I was also unclear about. Specifically, a more detailed explanation of the description length, to better understand what it is calculating/estimating and what information it is capturing from the gene network. Also, Reviewer 2 and 3 are uncertain about the choice of threshold for removing edges when the graph is reduced. What is considered a "viable set of genes" and why? Does removing these large p-value (with respect to correlation) edges have an effect on the results? Reviewer 2 suggests a small-scale study to elucidate these effects.

Model comparisons. It is difficult to tell if the comparisons made are appropriate because different gene network clusters were compared without evidence that it would not effect the results. All reviewers rightly flagged this issue. Wouldn't it make more sense to use the same reduced gene network with all models? And to include simulated gene networks (maybe several with distinct, but biologically realistic network topologies). to compare models? These comparisons using the same network for both (or all) models would be more convincing evidence if they showed that SBM captured something that WGCNA did not. Using more than one gene network type for comparison may also help readers understand the different instances where SBM excels or not compared to other models. Reviewer 1 gives more explanation on the use of simulated data, which I strongly suggest the authors consider.

Beyond these main issues the reviewers have other minor issues worth considering.

We cannot make any decision about publication until we have seen the revised manuscript and your response to the reviewers' comments. Your revised manuscript is also likely to be sent to reviewers for further evaluation.

Sincerely,

Jobran Chebib

Guest Editor

PLOS Computational Biology

Mark Alber

Section Editor

PLOS Computational Biology

Reviewer's Responses to Questions

**Comments to the Authors:**

Reviewer #1: Understanding the modularity or integration of organisms is of fundamental importance in evolutionary genetics, at the level of phenotypes such as morphology and life-history and of gene expression (which can also be considered a phenotype). To understand these patterns on an organismal scale necessitates the use of big data such as gene expression data, but dealing with these data in a way that is interpretable presents a host of challenges. Often dimension reduction techniques must be used. Careful consideration of the assumptions in such models and how those assumptions can bias the biological interpretation of the results is necessary.

In this manuscript the authors explore the utility of using a weighted hierarchical stochastic block model to analyse networks of gene co-expression and identify biologically meaningful clusters of genes. Although there are several different methods that can be used to do this, most impose an a-priori modular structure on the network, which could bias the subsequent inference of the modularity of gene co-expression networks. The purported benefit of the SBM is that no structure or tuning parameters are imposed on the data, such that any type of network structure is possible. If this is the case, estimates of modularity or integration, and the biological interpretation of the blocks of co-expressed genes should be less biased.

The idea of this paper is great. However, the fundamental differences between this type of model and existing approaches were not explained well-enough to understand any costs of this approach. Although it imposes no structure on the data, presumably there are some features of the data, which are not necessarily biological, that would be more likely to lead to certain outcomes than others.

The introduction of the SBM was quite vague and generic. For example, the statement on line 36-40 that SBM can use any information contained in the gene co-expression network while other methods can’t is unclear. All methods appear to be limited to the same information (the pairwise correlation between gene expression for every pair of genes in a network).

Line 27-28: does similar in this sentence mean more correlated? If so, it seems circular

It would be helpful to focus on introducing the conceptual differences between the SBM and other methods as they specifically relate to gene expression data, and any intricacies that go along with these types of data. A grounding in real biological examples would be useful as well. For example, Line 40- what other types of network organisations are expected in gene co-expression data?

A key aspect of this method that allows it to remain parameter free seems to be based on minimizing the description length. If this is the case, it would be very helpful if a more detailed explanation of the description length was given, and how it is estimated/calculated. What features of the data other than real biological signal could affect the minimum description length or choice of model. For example, if correlations of larger magnitude disproportionately contribute, are those high correlations more likely to be estimated in more highly expressed genes because of higher power, are they more likely to be estimated with less error, etc?

A major drawback of this manuscript is that the method has not been validated with simulated gene expression data. The authors showed that the different methods give different answers (although also largely congruent answers), but which one is correct? Analysing simulated data would also allow the authors to explore how different features of the data (eg. Mean expression level, estimation error, etc) influence the results.

The authors compare the results of the proposed SBM method with one other method (WGCNA) using empirical data from D. melanogaster. They attempted to compare with a third method (MMC), but note that the third method performed poorly with their data, so it is not really discussed. The authors should either discuss why the third method performed poorly and treat it like a real comparison, or remove it from the manuscript altogether.

Different edge weights were used for WGCNA and SBM which seems like a potentially important difference that could affect the results. For a true comparison the same gene co-expression network should be used for all methods even if it is possible to fit WGCNA and other methods with a denser network.

Overall, the manuscript is interesting and potentially important, but would benefit from 1) a clearer and more biologically grounded introduction; 2) validation of the approach using simulated data; 3) apples-to-apples comparison of the methods on empirical data.

Reviewer #2: In this manuscript, Melo et al use the Weighted Stochastic Block Model (SBM) to cluster genes in expression data from Drosophila melanogaster. They compare results from SBM with results from the commonly-used WCGNA method and show that SBM identifies more modular clusters with less assortativity than WCGNA, instead capturing other notions of similarity. Overall the paper is clear and well-presented and the claims made are justified by the results. I have a small number of questions and comments I would like to see addressed in a revision.

- Speaking generally, I am convinced that going beyond assortativity is desirable, especially given the inherently circular logic used in such algorithms. However I don’t end up with a great intuition for what exactly is driving the clustering if it isnt assortativity. This ends up coming through a bit in the methods and discussion eg “the amount of information required to encode the network”. But this remains a bit abstract. It would be helpful to get some intuition for what this is capturing at the end of the introduction.

- On that note, if it is indeed harder to get an idea of what exactly relates the genes in the identified clusters, this may be a downside to the method that should be mentioned in the discussion.

I am curious to hear more about the inclusion criteria for the genes and edges in the analysis:

- I’m not seeing a clear number of connections that can be included before you start running into computational limitations - eg the discussion around “We chose an FDR of 1% for the head and 0.1% for the body datasets which kept most of the genes while reducing the graph density to a manageable level for use in the SBM.“ This gives the number of genes but presumably the runtime increases with the number of edges. How many edges are you including and what is the runtime? What does the scaling look like with the number of edges?

- I am very curious about the information content of the removed edges, especially given you are using an inclusion criteria based on correlation, but a community detection method that is not. Is it possible to do a small-scale study on a smaller number of genes where you vary the FDR threshold eg from not removing anything down to a ~5% threshold? Would the major modules you find be the same? etc. If the communities detected are strongly influenced by this inclusion criteria, that would be a limitation of the analysis worthy of noting.

- I am also curious about the decision to correct for unknown covariates. Of course, this is a standard preprocessing step, but the way these methods typically work would suggest they may be likely to remove some large network structure prior to network analysis. I realize there isn’t really a “right” choice to be made here, but I am curious if the authors considered what the results would look like without this correction. For example, would some of the identified modules lack biological interpretability and thus be more likely to represent unobserved covariates?

Reviewer #3: The manuscript introduces an important discussion about the nature of network communities/clusters/blocks in gene networks. Specifically that there may be important clusters that do not follow the commonly held assumption of being assortative and that adhering to strongly to this assumption may negatively restrict our findings when analysing such networks.

I think this is an important argument to make, but there are technical issues with the current version of the manuscript that risk to undermine this message.

1. The whole premise is based on a comparison between modular methods on one network and the SBM on a different network. This does not really create a fair comparison since both the network and the method change. One solution might be to add another method: apply the Louvain method (or alternative modularity maximisation method) to the same network as used for the SBM.

2. Thresholded correlation matrices do not make good network representations. The authors acknowledge this in the discussion and point to more principled methods of reconstruction, so why not use them to construct the network? I understand that the approach used by the authors may be standard practice, but it is relatively straightforward to apply the graphical LASSO to a covariance matrix using available software (e.g., in python/R).

3. The block-wise measure of assortativity is presented poorly. (i) assortativity and modularity are related, but distinct statistics (see Newman 2003). (ii) The presented min and max values are wrong: +1 is not possible and the minimum is -0.5 (e.g. bipartite graph) (min and max values of assortativity are not trivial, see Cinelli et al 2020). (iii) The values of the block assortativity proposed depend on the assortativity of the other blocks, so the values might not have the interpretation the authors think they have. There are other methods for examining the assortativity at finer resolutions than the whole network (e.g., Peel et al 2018 + Cantwell and Newman 2019) It might be better to use the ratio of in/out edges or a heatmap of the (normalised) e_rs - e_r^2 matrix (or both).

Some minor points:

- Clustering all genes could be considered a negative as much as a positive given that most clustering algorithms will assign all input entities to a cluster.

- "Level-1 block 12 can also be referred to as 12-7-2-2-1 (see fig. 1" -- this does not correspond to anything in fig 1

- The parameters used to threshold the correlation matrix seem arbitrary. It would be good to have some justification and/or robustness check.

Cantwell, G. T., & Newman, M. E. J. (2019). Mixing patterns and individual differences in networks. _Physical Review E_, _99_(4), 042306.

Cinelli, M., Peel, L., Iovanella, A., & Delvenne, J. C. (2020). Network constraints on the mixing patterns of binary node metadata. _Physical Review E_, _102_(6), 062310.

Newman, M. E. (2003). Mixing patterns in networks. _Physical review E_, _67_(2), 02612

---

## [Decision Letter · Decision Letter 1]

7 Jul 2024

Dear Diego Melo et al.,

We are pleased to inform you that your manuscript 'Reassessing the modularity of gene co-expression networks using the Stochastic Block Model' has been provisionally accepted for publication in PLOS Computational Biology. We apologize for the delay in finding appropriate reviewers to review this work. There are only a few minor considerations from one reviewer that we would suggest including in the final submission.

Best regards,

Jobran Chebib

Guest Editor

PLOS Computational Biology

Pedro Mendes

Section Editor

PLOS Computational Biology

Reviewer's Responses to Questions

**Comments to the Authors:**

Reviewer #2: The authors have produced a nice revision that addresses my concerns.

Reviewer #4: the review is uploaded as an attachment

**Have the authors made all data and (if applicable) computational code underlying the findings in their manuscript fully available?**

Reviewer #2: Yes

Reviewer #4: Yes

PLOS authors have the option to publish the peer review history of their article (what does this mean?). If published, this will include your full peer review and any attached files.

Reviewer #2: No

Reviewer #4: **Yes: **Francesco Zambelli

---

## [Editor Report · Acceptance letter]

23 Jul 2024

PCOMPBIOL-D-23-01919R1 

Reassessing the modularity of gene co-expression networks using the Stochastic Block Model

Dear Dr Melo,

I am pleased to inform you that your manuscript has been formally accepted for publication in PLOS Computational Biology. Your manuscript is now with our production department and you will be notified of the publication date in due course.

With kind regards,

Zsofia Freund
